# Single Amino Acids as Sole Nitrogen Source for the Production of Lipids and Coenzyme Q by *Thraustochytrium* sp. RT2316-16

**DOI:** 10.3390/microorganisms12071428

**Published:** 2024-07-14

**Authors:** Liset Flores, Carolina Shene

**Affiliations:** Department of Chemical Engineering, Center of Food Biotechnology and Bioseparations, BIOREN, and Centre of Biotechnology and Bioengineering (CeBiB), Universidad de La Frontera, Temuco 4780000, Chile; liset.flores@ufrontera.cl

**Keywords:** amino acids, metabolism, thraustochytrids, *Thraustochytrium* sp., vitamins

## Abstract

This work analyzes the production of total lipids and the content of CoQ_9_ and CoQ_10_ in the biomass of *Thraustochytrium* sp. RT2316-16 grown in media containing a single amino acid at a concentration of 1 g L^−1^ as the sole nitrogen source; glucose (5 g L^−1^) was used as the carbon source. Biomass concentration and the content of total lipids and CoQ were determined as a function of the incubation time; ten amino acids were evaluated. The final concentration of the total biomass was found to be between 2.2 ± 0.1 (aspartate) and 3.9 ± 0.1 g L^−1^ (glutamate). The biomass grown in media containing glutamate, serine or phenylalanine reached a content of total lipids higher than 20% of the cell dry weight (DW) after 72, 60 and 72 h of incubation, respectively. The highest contents of CoQ_9_ (39.0 ± 0.7 µg g^−1^ DW) and CoQ_10_ (167.4 ± 3.4 mg g^−1^ DW) in the biomass of the thraustochytrid were obtained when glutamate and cysteine were used as the nitrogen source, respectively. Fatty acid oxidation, which decreased the total lipid content during the first 12 h of incubation, and the oxidation of hydrogen sulfide when cysteine was the nitrogen source, might be related to the content of CoQ_10_ in the biomass of the thraustochytrid.

## 1. Introduction

Coenzyme Q (CoQ), also known as ubiquinone, is a lipophilic molecule that consists of a fully substituted benzoquinone ring to which to a polyisoprenic tail is attached. In addition to its fundamental role as an electron carrier, CoQ is involved in pyrimidine biosynthesis, mitophagy, and the regulation of sulfide metabolism [1]. In its reduced form (CoQH_2_), CoQ is the only lipid-soluble antioxidant synthesized endogenously. CoQ functions as a membrane-localized antioxidant protecting cells against lipid peroxidation [2]. In addition, CoQ is required as an electron acceptor in one of the dehydrogenase reactions of the β-oxidation cycle [3]; CoQ regulates the opening of the mitochondrial potential transition pore [4], and is also required by the uncoupling protein UCP1 that acts as a proton translocator, allowing the dissipation of the proton gradient arising from the functioning of the respiratory chain [5].

The length of the polyisoprenic tail, which keeps CoQ anchored to cell membranes, is determined by the number of isoprenes and varies depending on the species; *Escherichia coli*, *Saccharomyces cerevisiae* and *Caenorhabditis elegans* produce CoQ with eight (CoQ_8_), six (CoQ_6_), and nine (CoQ_9_) isoprene units, respectively. *Schizosaccharomyces pombe* and *Homo sapiens* have the CoQ with the longest length (CoQ_10_). Some organisms synthesize CoQs with different tail lengths, as is the case of rodents (CoQ_9_ and CoQ_10_) and humans (CoQ_9_ and CoQ_10_).

In recent years, a great interest in CoQ_10_ has been generated due to its health benefits, specifically in the treatment of various diseases such as cardiovascular, neurodegenerative, and mitochondrial diseases [6]. CoQ_10_ is used in the treatment of some types of cancer (especially breast cancer) and hypertension [7]. In addition, CoQ_10_ is a potent antioxidant that protects cells from free radical damage [8]. These properties have promoted its use by the pharmaceutical and cosmetic industry [9]. The microbial production of CoQ_10_ could be an alternative to chemical synthesis to satisfy the growing demand of the industry [10]. Different microorganisms capable of producing CoQ_10_ have been reported: *Agrobacterium tumefaciens* [11], *Rhodobacter sphaeroides* [12], *Paracoccus denitrificans* [13], *Sphingomonas* [14], *Proteus* [15], recombinant *Escherichia coli* [16,17], *Candida* [18] and *Rhodotorula* [19]. A novel source of CoQ_10_ is *Thraustochytrium* sp. RT2316-16 [20].

Thraustochytrids are heterotrophic, unicellular eukaryote microorganisms, found ubiquitously in the marine environment. The order Thraustochytriales [21,22] was established for protists characterized by: a monocentric thallus; an ectoplasmic net system with an associated sagenogenetosome; a mutilamellate wall composed of dictyosome-derived scales; and biflagellate, heterokont zoospores [23]. These protists are primary decomposers, playing a significant role in nutrient recycling [24]. Certain marine thraustochytrids are used for the commercial production of the long-chain omega-3 polyunsaturated fatty acid, docosahexaenoic acid [25]. Under suitable culture conditions, the lipid content of thraustochytrids can exceed 55% of the dry cell weight (DW) [25,26,27]. Thraustochytrids have also been studied for the production of bioactive compounds of commercial interest such as squalene, carotenoids, phospholipids and exopolysaccharides [28,29,30].

The potential of each new thraustochytrid for use in the synthesis of lipidic compounds of industrial interest is generally evaluated in growth media formulated with complex organic nitrogen sources (yeast extract and/or peptones). The compositions of these sources can vary between batches; also, these sources provide other nutrients that in some cases are unknown. For the production of certain cell compounds (nutraceutical and pharmaceutical), it is desirable to use a chemically defined medium. These media allow for the better control of the culture process, can facilitate downstream processing, and yield products with a constant quality [31]. Previous studies conducted with a single amino acid used as the nitrogen source [32] or an amino acid in combination with ammonium sulfate [33] allowed for defining which amino acids promote the growth of *Thraustochytrium* sp. RT2316-16. Based on these results, ten amino acids were selected to study their effects on the content of lipids and the two coenzymes Q (CoQ_9_ and CoQ_10_) in the biomass of RT2316-16. The results were explained by relating the metabolisms of the different amino acids to the synthesis/consumption of total lipids. Kinetic parameters estimated from the experimental data were also used to explain the results.

## 2. Materials and Methods

### 2.1. Microorganism and Inoculum Preparation

The strain RT2316-16, similar to *Thraustochytrium* sp. (Accession Number NCBI MT648462), was used in all the experiments. Pure stock cultures were kept frozen at −80 °C in 50% (v v^–1^) glycerol–artificial seawater (ASW). ASW contained the following per liter of distilled water: NaCl 27.5 g, MgCl_2_·6H_2_O 5.38 g, MgSO_4_·7H_2_O 6.78 g, KCl 0.72 g, NaHCO_3_ 0.2 g, and CaCl_2_·2H_2_O 1.6 g.

The inoculum for the experiments was grown aseptically in Erlenmeyer flasks (250 mL) containing 100 mL of a sterile medium of the following composition (g L^–1^): glucose (Merck, Darmstadt, Germany) 20, yeast extract (Merck, Darmstadt, Germany) 6, and monosodium glutamate (Merck) 0.6; the medium was made using a 1: 1 (v v^–1^) mixture of distilled water and ASW. A trace elements solution (24 mL L^–1^) and two vitamin solutions (vitamin solution I, 3.6 mL L^–1^; vitamin solution II, 3.6 mL L^–1^) were filter-sterilized by passing them through a 0.2 µm sterile membrane filter and added to the medium. The trace elements solution contained the following in distilled water (g L^–1^): MnCl_2_·4H_2_O 0.3, ZnSO_4_·7H_2_O 0.3, CoCl_2_·6H_2_O 0.004, CuSO_4_·5H_2_O 0.2, NiSO_4_·6H_2_O 0.2, FeSO_4_·7H_2_O 1, and KH_2_PO_4_ 5 [34]. Vitamin solution I contained the following in distilled water (g L^–1^): thiamine (vitamin B1) 0.04, Ca-pantothenate (B5) 0.02, nicotinic acid (B3) 0.02, and pyridoxine (B6) 0.008 (Sigma-Aldrich, St Louis, MO, USA) [34]. Vitamin solution II contained the following in distilled water (g L^–1^): biotin (B7) 0.01, cobalamin (B12) 0.001, riboflavin (B2) 0.1, pyridoxamine (B6) 0.2, and *p*-aminobenzoic acid 0.02 (Sigma-Aldrich, St Louis, MO, USA) [34]. Here, 1 mL of a thawed stock culture was added to 100 mL of the above medium and the flask was incubated (15 °C) aseptically on an orbital shaker (150 rpm) for 4 days. To ensure the thraustochytrid was active, a second inoculum prepared exactly as the first was inoculated with 5 mL of the first inoculum and incubated under the same conditions. This second inoculum was used in the growth experiments.

### 2.2. Effect of Culture Medium’s Composition on the Content of Total Lipids and CoQ_10_ in the Biomass of RT2316-16

The composition of the control medium used in these experiments was (g L^−1^): glucose 5, yeast extract 6, monosodium glutamate 0.6, and a yeast nitrogen base without amino acids and ammonium (YNB) (Becton, Dickinson and Company, Sparks, MD, USA) 6.7. The required amount of YNB was dissolved in sterile distilled water and filter-sterilized by passing through a 0.2 µm sterile membrane filter. The medium had a final ASW concentration equal to 50% v v^−1^. Erlenmeyer flasks (250 mL) containing 100 mL of the sterile medium were inoculated with the cells in 5 mL of a second inoculum that were centrifuged (6000× *g*, 4 °C, 10 min), washed (5 mL of sterile and diluted ASW), centrifuged again, and suspended in 5 mL of sterile ASW (50% v v^−1^). Twenty-four flasks were inoculated to obtain the growth curve. Every 12 h, 3 flasks were withdrawn; the content was centrifuged (6000× *g*, 4 °C, 10 min) and the cell pellet was washed with distilled water, lyophilized, weighed and stored at −20 °C for lipid and CoQ analysis; the cell-free medium was stored at −20 °C for analysis.

In the first experiment, the effect of the inoculum was tested; 24 flasks containing 100 mL of the sterile control medium were inoculated with 5 mL of the grown culture without any treatment. In the second experiment the effects of vitamins and trace minerals were tested. To do this, the control medium was supplemented with the trace elements solution (24 mL L^–1^), and the two vitamin solutions (vitamin solution I, 3.6 mL L^–1^; vitamin solution II, 3.6 mL L^–1^). In the third experiment, the medium had a yeast extract concentration equal to 1 g L^−1^.

To test the effect of a single amino acid as the sole nitrogen source, the composition of culture medium was (g L^−1^): glucose 5, YNB 6.7, and the amino acid (cysteine (Cys), lysine (Lys), glutamine (Gln), alanine (Ala), glutamate (Glu), aspartic acid (Asp), serine (Ser), leucine (Leu), phenylalanine (Phe), or proline (Pro)) 1. Proline was also tested at 2 g L^−1^. The pH of the medium containing aspartic acid was 5.3 and did not allow the growth of RT2316-16. Thus, the pH of this medium was brought to 7 with NaOH (2 M) before sterilization.

### 2.3. Analyses

#### 2.3.1. Cell Dry Weight and the Concentration of Lipid Free Biomass

The concentration of dry biomass (*xt*) was determined gravimetrically; a known culture volume (10 mL) was centrifuged (2057× *g*, 10 min), and the cell pellet was washed twice with distilled water, recovered by centrifugation, and dried to constant weight at 105 °C. The concentration of lipid-free biomass (*xlf*) was calculated using:(1)$$xlf=xt\left(1-\frac{l}{100}\right)$$
where *l* is the total lipid content of the biomass (percentage of total DW). The standard deviation of *xlf* was calculated by the propagation of errors [35].

#### 2.3.2. Residual Reducing Sugars

The residual glucose concentration in the cell-free medium was measured spectrophotometrically using the 3.5-dinitrosalicylic acid (DNS) method; the DNS reagent was prepared as described by Zhang et al. [36]. The calibration curve was built using glucose solutions with concentrations in the range 0 to 1 g L^−1^.

#### 2.3.3. Extraction and Quantification of CoQ

The extraction of CoQ from the biomass was performed using the methodology described by Martínez et al. [37] with some modifications. A 50 mg sample of the freeze-dried powdered biomass was suspended in 6 mL of cold methanol (−20 °C) and sonicated (20 min, 4 °C). Petroleum ether (6 mL) was added and the suspension was vortex mixed (1 min) and centrifuged (4000× *g*, 4 °C, 10 min). The upper phase was extracted twice with petroleum ether (3 mL per extraction) and the extracts were pooled in a vial. The solvent in the pooled extract was evaporated under a stream of nitrogen. The residue was suspended in 450 µL of a solvent mixture (methanol: ethanol = 65:35 v v^−1^) and 50 µL of a ferric chloride (FeCl_3_) solution (1% w v^−1^ in ethanol) was added to reduce CoQ to CoQH_2_. The sample was filtered with a 0.22 µm syringe filter before HPLC analysis.

For the HPLC analysis, a C18 column (250 × 4.6 mm, 5 µm; Symmetry C18, Waters, Milford, MA, USA) kept at 30 °C was used. Detection was performed at 275 nm (Waters 2487 dual detector). The mobile phase comprised a mixture of methanol and ethanol (65:35 v v^−1^) at a flow rate of 1.0 mL min^−1^ [38]. The injection volume was 20 µL. Standard solutions of CoQ_10_ and CoQ_9_ (Sigma-Aldrich) reduced with ferric chloride were used for identification and quantification.

### 2.4. Determination of Kinetic Parameters and Statistical Analysis

Kinetic parameters were estimated using the experimental data as described in Appendix A.

The statistical significance of the difference between the growth curve (concentrations of total biomass and lipid-free biomass as function of incubation time) of the tests and the respective control curve was determined using the method proposed by Hristova and Wimley [39]. Curves of the contents of lipid and the two coenzymes Q in the biomass were also compared with the respective control and analyzed for the statistical significance of the differences.

## 3. Results

### 3.1. Effect of the Inoculum, Vitamins and Trace Minerals, and Yeast Extract Concentration on the Growth and the Content of Lipids in Thraustochytrium *sp.* RT2316-16

To test the effects of the medium composition on cell growth and the synthesis of metabolites in cases where a different medium is used to grow the inoculum, residual components (nutrients and extracellular metabolites) in the inoculum must be eliminated or reduced. One way to do this is to discard the supernatant after centrifugation and wash the cell pellet. Figure 1a compares the growth curves obtained with an inoculum consisting of the cells in the aliquot of the grown culture (5% v v^−1^) that were washed (the control experiment), or the aliquot from the same culture without any treatment (cells plus the residual medium). Growth curves of total biomass and lipid-free biomass (Equation (1)) are presented to identify increases in the biomass concentration due to lipid accumulation.

The treatment of the inoculum affected the specific growth rate and the rate of glucose consumption (Appendix A). Nonetheless, the differences between the growth curves of lipid-free biomass obtained with the two inocula were not statistically significant (*p* > 0.05), and the final concentration of the lipid-free biomass was the same (4.4 ± 0.3 g L^−1^) (Figure 1a). During the incubation period, the total lipid content of the biomass presented important changes (Appendix A); a decrease in the total lipid content (from 22 to 12% DW) was observed during the first 12 h; after 12 h the total lipid content of the biomass increased to a level close to the initial content. The inoculum had a significant effect (*p* < 0.05) on the evolution of the content of the two coenzymes Q in the biomass; the treatment reduced the contents of CoQ_9_ and CoQ_10_ in the biomass of RT2316-16, an effect observed between 12 and 72 h and between 12 and 48 h, respectively (Figure 1b). The contents of the two co-enzymes Q in the biomass showed a different behavior as a function of the incubation time; while the content of CoQ_9_ decreased after the inoculation (0–12 h), increasing then from 12.7 ± 1.9 µg g^−1^ at 12 h to 55.1 ± 1.5 µg g^−1^ at 96 h, the content of CoQ_10_ decreased (from the initial 14.7 ± 1.7 mg g^−1^ to zero after 60 h).

The effect of the initial yeast extract concentration was tested at 1 g L^−1^ (low concentration) and 6 g L^−1^ (high concentration, the control) with the same initial glucose concentration (5 g L^−1^). The growth curves (total biomass and lipid-free biomass) obtained with the low and high concentration of yeast extract were statistically different (*p* < 0.05); this was also the case for the curves of the lipid content. The low yeast extract concentration did not allow the biomass to grow during the first 12 h, and the final concentrations of total biomass and lipid-free biomass were lower than the concentration obtained with the high yeast extract concentration (5.5 ± 0.1 and 4.4 ± 0.1 g L^−1^, respectively) (Figure 1c). All this notwithstanding, glucose concentration decreased at the same rate (Figure 1c). The initial yeast extract concentration had a significant effect (*p* < 0.05) on the content of the two coenzymes Q in the biomass of RT2316-16 during the incubation. The final CoQ_9_ content in the biomass grown with the high concentration of yeast extract was 1.6-fold higher than the content in the biomass grown with the low yeast extract concentration (34.7 ± 1.1 µg g^−1^) (Figure 1d). The CoQ_10_ content in the biomass grown with the low yeast extract concentration decreased to almost zero during the first 12 h; the increase in the CoQ_10_ content observed after 48 h (Figure 1d) occurred 12 h before the specific growth rate of the total biomass showed a significant decrease (0.033 h^−1^ at 60 to 0.004 h^−1^ at 72 h) (Appendix A).

The culture medium used in the above-described experiments contained the vitamins and trace minerals in YNB, a product that is used to test the effect of nitrogen and carbon sources on the growth of yeasts. The effect of supplementing the medium with the vitamins and trace minerals solutions used in the isolation and culture of thraustochytrids (composition and dose described in Section 2) was tested. A comparison of the composition of the vitamins and trace minerals in the two media is shown in Appendix A. The extra vitamins and trace minerals had a significant effect (*p* < 0.05) on the growth curves of total biomass and the lipid-free biomass (Figure 1e), although the final concentrations were not different. Vitamins and trace minerals supplementation had no significant effect (*p* > 0.05) on the evolution of total lipid content of the biomass as a function of the incubation time; the highest contents of lipids were 26.6 ± 0.5% (84 h) and 25.9 ± 0.8% (72 h) in the biomass grown with the supplemented and control media, respectively (Appendix A). The extra vitamins and trace minerals had different effects on the content of the two coenzymes Q in the biomass; compared with the control, the evolution of the content of CoQ_10_ in the biomass grown in the supplemented medium showed no significant differences (*p* > 0.05), whereas the differences in the content of CoQ_9_ were significant (*p* < 0.05) (Figure 1f).

### 3.2. Effect of a Single Amino Acid as Nitrogen Source on the Growth and Lipid Content of Thraustochytrium *sp.* RT2316-16

Growth curves of *Thraustochytrium* sp. RT2316-16 in media containing glucose at 5 g L^−1^ and a single amino acid as the sole nitrogen source (1 g L^−1^) were obtained. The total lipid content and the contents of the two coenzymes Q in the biomass were determined. Amino acids used to grow the thraustochytrid were glutamine, glutamate, phenylalanine, proline, lysine, alanine, aspartate, leucine, serine and cysteine. The effect of proline was also tested at 2 g L^−1^. The graphs in Figure 2 show concentrations of total biomass, lipid-free biomass and glucose during the incubation period. The total lipid content in the biomass harvested at the different times is shown in Appendix A. An important decrease in the total lipid content during the first 12 h incubation that depended on the amino acid was observed. The growth of total biomass stopped (at different times) in the cultures made with glutamine (24–36 h), lysine (12–24 h) and proline (1 g L^−1^) (24–36 h), and then resumed. The final concentration of the total biomass was found between 2.2 ± 0.1 (aspartate) and 3.9 ± 0.1 g L^−1^ (glutamate). The contents of total lipids in the biomass increased more than 20% of DW when glutamate, serine or phenylalanine were used as the nitrogen source (Appendix A). Glucose was exhausted in the cultures with cysteine and leucine, whereas the lowest fraction of consumed glucose was obtained with proline; the fractions of consumed glucose were 35 and 56% for proline concentrations of 1 and 2 g L^−1^, respectively. Each of the growth curves obtained with the different amino acids were compared with the growth curve obtained with yeast extract at 1 g L^−1^, used as the control. The differences between the growth curve of the control and the curve obtained with alanine or proline (both concentrations) were not significant (*p* > 0.05). Also, the differences between the curves of the total lipid contents of the biomass grown with alanine and the control were not significant (*p* > 0.05); this was also the case of the lipid content in the biomass grown with glutamate.

Figure 3a shows the mass ratio of the CoQ_9_ content and the initial content (CoQ_9,0_) in the total biomass at the different the incubation times. The comparison of the curves CoQ_9_ as a function of incubation time obtained with the alanine, proline (both concentrations), or leucine and the control (Figure 1d) shows that differences were not significant (*p* > 0.05). After 96 h, the biomass grown with glutamate and aspartate attained the highest (39.0 ± 0.7 µg g^−1^ DW) and the lowest (21.8 ± 0.4 µg g^−1^ DW) contents of CoQ_9_, respectively. The graph in Figure 3b shows the mass ratio of the CoQ_10_ content and the initial content (CoQ_10,0_) in the total biomass for the different incubation times. The curves of CoQ_10_ content in the biomass as a function of incubation time obtained with all the amino acids were significantly different (*p* > 0.05) from the control curve (Figure 1d). Glutamine as the only nitrogen source did not promote the accumulation of CoQ_10_ in the biomass, whereas CoQ_10_ was found in the biomass grown with cysteine, alanine, serine and leucine at the different times. After 12 h, the biomass grown with cysteine had the highest content of CoQ_10_ (167.4 ± 3.4 mg g^−1^ DW), followed by the contents in the biomass grown with aspartate (96.9 ± 2.1 mg g^−1^ DW), alanine (63.4 ± 1.5 mg g^−1^ DW), serine (31.2 ± 1.6 mg g^−1^ DW), leucine (24.1 ± 4.9 mg g^−1^ DW), and proline (1 g L^−1^) (20.8 ± 0.1 mg g^−1^ DW) (Figure 3b; Appendix A).

To determine if the contents of the two CoQ in the biomass of RT2316-16 after 12 h of incubation were related with some of the kinetic parameters of the culture (specific growth rate of the total biomass, specific glucose consumption rate, specific rate of lipid synthesis/consumption shown in Appendix A), the correlation coefficients of the different linear regressions were determined. The linear relationships with the highest correlation were seen for the content of CoQ_9_ in the biomass in terms of the specific lipid synthesis/consumption (r_L_) (r = 0.951; R^2^ = 0.905) (Figure 4a), and for the content of CoQ_10_ in terms of the specific growth rate of the total biomass (r_X_) (r = 0.731; R^2^ = 0.535) (Figure 4b).

## 4. Discussion

Co-enzyme Q is an electron carrier involved in ATP synthesis and many other reactions important for cell growth. Thus, it is of interest to study how culture conditions affect its synthesis in microorganisms able to produce CoQ_10_, the main form in human cells, especially if it is considered that microbial CoQ_10_ has the potential to be used as a dietary supplement for the treatment of some deceases [40]. Flores et al. reported that *Thraustochytrium* sp. RT2316-16 has the ability to produce CoQ_9_ and CoQ_10_ [20]; CoQ_9_ was detected in cells grown in different culture media, suggesting its involvement in the electron transfer chain. The conditions that trigger the accumulation of CoQ_10_ and its cell function in the thraustochytrid remain to be determined.

Because cell growth and cell composition are determined by the composition of the growth medium, this work analyzed the effect of a single amino acid used as the sole nitrogen source for growing RT2316-16 on the content of total lipids and CoQ in the biomass. The best nitrogen source for the growth of thraustochytrids is generally yeast extract, which provides peptides and free amino acids. Amino acids can be used directly in protein synthesis and their carbon skeletons can be used to produce fatty acids.

Yeast extract promoted the growth of the lipid-free biomass of RT2316-16 and the simultaneous accumulation of total lipids to a level that depended on its concentration (Figure 1c and Appendix A). In a previous work, it was reported that the total biomass concentration of RT2316-16 increased more than 2-fold after 48 h when proline, lysine, glutamine, serine, leucine, phenylalanine, isoleucine, tryptophane, cysteine, alanine, or aspartic acid were used as the only source of nitrogen [32]. The results in this work show that RT2316-16 was able to grow on seven (proline, lysine, glutamine, glutamate, serine, leucine and phenylalanine) of the ten amino acids evaluated to a level similar to or better than yeast extract at the same concentration (1 g L^−1^). Cysteine, alanine, and aspartate also allowed the growth of RT2316-16, but the final concentration of the total biomass was lower than the concentration obtained with yeast extract (Figure 2). The results also show that the consumption of glucose by RT2316-16 was determined by the amino acid, and only two of them (cysteine and leucine) allowed its exhaustion after 96 h, which was 24 h later than when yeast extract was used.

In the batch culture experiments, the total lipid content of the thraustochytrid biomass changed with the incubation time (Appendix A); after 12 h of incubation, the initial content of total lipids decreased 41% in the serine medium and 71% in the leucine medium. The changes in total lipid content of the biomass during the subsequent incubation period (12–96 h) were determined by the residual glucose concentration, the initial concentration of yeast extract, or the amino acid used as nitrogen source (Appendix A). Glutamate, phenylalanine, alanine, and serine allowed to attain a lipid content (21.0, 20.5, 18.9, and 21.2% DW, respectively Appendix A; Appendix A) higher than the lipid content of the biomass grown with the yeast extract at a concentration of 1 g L^−1^ (17.9% DW; Appendix A). Acetyl-CoA and reducing power for the synthesis of fatty acids, the components of total lipids, are produced in the metabolism of glucose. However, amino acids can also contribute to fatty acid synthesis either by limiting the growth of the lipid-free biomass, allowing the carbon to be channeled to fatty acid synthesis, or/and by being the source of carbon for fatty acid synthesis (Figure 5).

Fatty acids are synthesized in the cytosol from acetyl-CoA formed in the reaction catalyzed by ATP citrate lyase (ACLY; Figure 5); citrate, the substrate of this reaction, has to be exported from the mitochondria to the cytosol. This occurs when the TCA cycle is inhibited by high concentrations of ATP and NADH. Citrate synthase (CS), the first reaction in the TCA cycle, produces citrate from oxaloacetate and acetyl-CoA, produced from malate or α-ketoglutarate, and pyruvate, respectively. Pyruvate, the end product of glycolysis, is also produced in the metabolism of alanine and serine. These two amino acids used as a nitrogen source produced a biomass with a high lipid content (Appendix A). Cysteine is another potential source of pyruvate (Figure 5); however, the use of cysteine as a nitrogen source did not promote the accumulation of total lipids in RT2316-16 (Appendix A). Glutamate that is metabolized to α-ketoglutarate promoted the growth of a biomass with a high lipid content (21% DW, 72 h). The metabolism of glutamine and proline produces glutamate; however, only glutamine allowed the accumulation of total lipids (18.3% DW, 48 h). The low lipid content (less than 6% DW) of the biomass grown with proline (1 g L^−1^) is explained by the low glucose uptake (Figure 2). To test if this effect was due to the concentration of proline, a culture with a higher concentration (2 g L^−1^) was made. A higher proline concentration (2 g L^−1^) did not increase glucose consumption but slightly increased the content of total lipids in the biomass (Appendix A). The oxidation of proline to glutamate involves the activity of proline dehydrogenase (ProDH), a FAD-dependent enzyme that produces delta1-pyrroline-5-carboxylate (P5C), and P5C dehydrogenase (P5CDH) that converts glutamate semialdehyde (GSA in equilibrium with P5C) to glutamate. ProDH transfers the electrons to the electron transfer flavoprotein ubiquinone oxidoreductase (ETF-QO) for the production of ATP (Figure 5); the second reaction reduces NAD^+^ or NADP^+^ [41]. The low consumption of glucose when proline was used as the sole nitrogen source could be explained by an impaired electron transfer chain due to reactive oxygen species formed by the activity of ProDH [42].

The metabolism of phenylalanine (through the tyrosine metabolic pathway) produces as end products fumarate and acetoacetate. In the TCA cycle, fumarate is hydrated to malate, the substrate of malate dehydrogenase that produces the oxaloacetate needed for the synthesis of acetyl-CoA. Acetoacetate can feed the TCA cycle after being transformed into two acetyl-CoA molecules through reactions catalyzed by succinyl-CoA:3-ketoacid CoA transferase (SCOT), which produces acetoacetyl-CoA, and acetoacetyl-CoA thiolase [43] (Figure 5). Leucine metabolism also produces acetoacetate; however, the lipid content of the biomass grown with phenylalanine was higher (20.5% DW at 72 h) than the lipid content of the biomass grown with leucine (13.3% DW at 60 h). It could be possible that the activity of isovaleryl-CoA dehydrogenase (IVDH) in the metabolism of leucine affects the availability of the acetyl-CoA needed for fatty acid synthesis; this is a flavoenzyme that catalyzes the conversion of isovaleryl-CoA to 3-methylcrotonyl-CoA and transfers electrons to ETF-QO via electron transferring flavoprotein (ETF). The metabolism of lysine also involves a flavoenzyme, glutaryl-CoA dehydrogenase (GCDH), that oxidatively decarboxylates glutaryl-CoA to crotonyl-CoA, which is further transformed into acetoacetyl-CoA. In contrast to phenylalanine, lysine did not allow the accumulation of total lipids in RT2316-16 (Appendix A).

The low concentration of total biomass and the low lipid content of this biomass in the culture with aspartate as the sole nitrogen are explained by the reduced consumption of glucose. Aspartate, the substrate for the synthesis of purine and pyrimidine nucleobases and other metabolites needed for cell proliferation, is synthesized in the mitochondria, from where it is unidirectionally exported in exchange with cytosolic glutamate plus a proton by the aspartate–glutamate carrier (AGC) [44]. AGC is a fundamental component of the malate–aspartate shuttle (MAS) a biochemical pathway used for transferring NADH from the cytosol to mitochondria along with the glycerol-3-phosphate shuttle [44]. A high cytosolic concentration of aspartate, when this amino acid was the sole nitrogen source, might have interfered with the activity of AGC and thus the transfer of NADH from cytosol to mitochondria, slowing glucose uptake and energy production and thus limiting the growth of the thraustochytrid. The inhibitory effect of aspartate on the activity of the anaplerotic enzymes phosphoenolpyruvate carboxylase [45] and pyruvate carboxylase [46] could also explain the reduced consumption of glucose in RT2316-16.

The experimental results show that the contents of the two co-enzymes Q in the biomass of RT2316-16 were highly dependent on the nitrogen source and the incubation time. After the significant decrease in the CoQ_9_ content in RT2316-16 during the initial 12 h (Appendix A), the CoQ_9_ content increased as the concentration of glucose decreased, while the content of CoQ_10_ showed the opposite trend (Figure 1 and Figure 4; Appendix A); the observed differences might suggest the involvement of CoQ_9_ and CoQ_10_ in different cell functions. Another possibility is that their synthesis is triggered by growth conditions that are different during the first 12 h of incubation.

During the first 12 h, the content of total lipids in the biomass decreased (Appendix A) due to dilution by cell growth or fatty acid (short- and medium-chain) degradation through mitochondrial β-oxidation. Reactions in the β-oxidation cycle catalyzed by acyl-CoA dehydrogenases (ACADs), flavoproteins [47], and 3-hydroxyacyl-CoA dehydrogenases (HADs) [48] produce FADH_2_ and NADH, respectively. In the first reaction, the two reducing equivalents generated are transferred to ETF, and from this to the respiratory chain via ETF dehydrogenase [47]. In complex I, the two electrons from NADH are transferred to ubiquinone (CoQ). The high content of CoQ_10_ in the 12 h biomass of RT2316-16 grown with most of the amino acids could be related to requirements for transferring the electrons produced during the β-oxidation of fatty acids. The absence of CoQ_10_ in the biomass of RT2316-16 grown with glutamine (through all the incubation period) and glutamate (the first 24 h) might be due to the lower decrease in total lipids content in the biomass (44 and 51%, respectively). The presumed involvement of CoQ_10_ in fatty acid oxidation might explain the relationship between the specific growth rate of the total biomass and the content of CoQ_10_ in the 12 h biomass (Figure 4b). Also, the contribution of lipid metabolization to the initial cell growth might explain the relationship between the content of CoQ_9_ in the biomass and the specific rate of lipid synthesis/consumption (Figure 4a).

In batch, cultures the specific growth rate decreases due to the decrease in the concentration of nutrients. Also, the increase in biomass concentration with time increases the oxygen uptake rate, decreasing the level of dissolved oxygen. It has been reported that a limited supply of oxygen, the final electron acceptor in the respiration chain, enhances CoQ_10_ production by *Agrobacterium tumefaciens* ATCC4452 [49] and *Rhizobium radiobacter* T6102 [50]. The authors proposed that a partial restriction of the electron flux under oxygen limiting conditions allows the accumulation of the reduced form of CoQ_10_ (CoQ_10_H_2_); to adapt the imbalance, more CoQ_10_ is synthesized. A similar mechanism might explain the increase in the CoQ_9_ content in RT2316-16 as the incubation time increased (Figure 1; Appendix A).

The very high content of CoQ_10_ in RT2316-16 grown with cysteine could be due to its role in the oxidation of hydrogen sulfide. Cysteine metabolism involves its dioxygenation to 3-sulfinoalanine in the reaction catalyzed by cysteine dioxygenase; the formed product is then transaminated with α-ketoglutarate by aspartate aminotransferases to yield glutamate and 3-sulfinylpyruvate that spontaneously decomposes to pyruvate and hydrogen sulfide (Figure 5) [51]. A sulfide quinone oxidoreductase (SQOR), a flavoprotein disulfide reductase that resides in the inner mitochondrial membrane in humans, catalyzes the two-electron oxidation of hydrogen sulfide to sulfane sulfur, transferring the electrons to CoQ and the sulfane sulfur to an acceptor substrate (possibly glutathione or sulfite) [52] (Figure 5). The other amino acids that promoted the accumulation of CoQ_10_ in RT2316-16 were aspartate, leucine and proline, which have in common a notable decrease in the total lipid content during the first 12 h (Appendix A).

## 5. Conclusions

*Thraustochytrium* sp. RT2316-16 can use glutamine, glutamate, phenylalanine, alanine or serine as the sole nitrogen source for growing and lipid accumulation (>20%). The content of CoQ_9_ in the biomass increases during the incubation period; proline and leucine allowed increases of CoQ_9_ in the biomass similar to those obtained with yeast extract. The content of CoQ_10_ in the biomass strongly depended on the amino acids used as the source of nitrogen. While this coenzyme was not detected in the biomass grown with glutamine, the biomass grown with cysteine had the highest content of CoQ_10_. The results obtained in this work could act as the base for defining a chemically defined medium for the growth of related thraustochytrids.

## Figures and Tables

**Figure 1 microorganisms-12-01428-f001:**
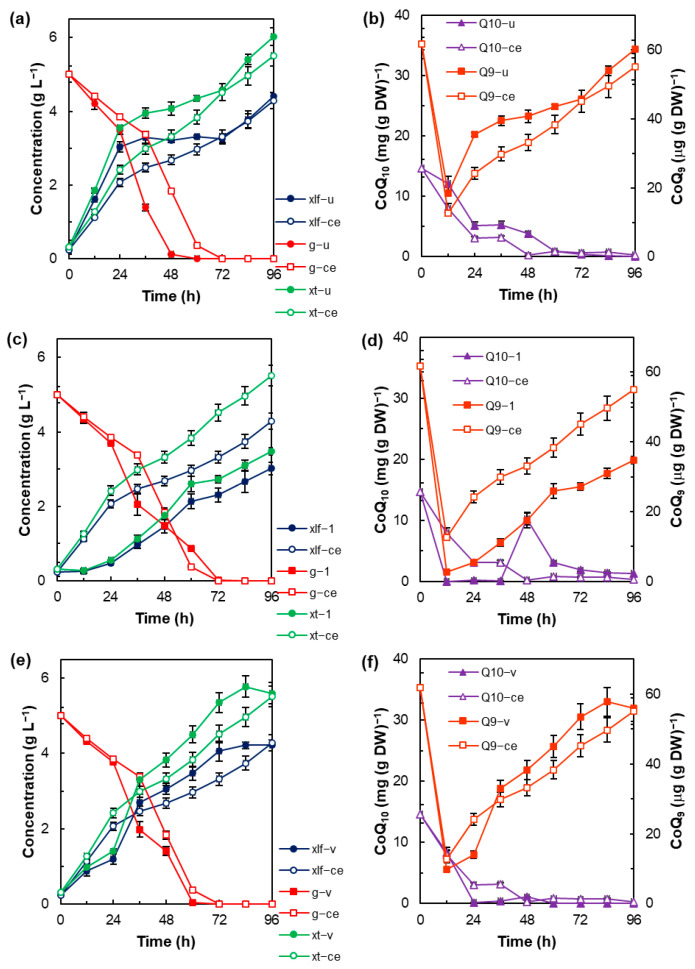
Effect of: the inoculum (**a**,**b**) (ce, the control experiment inoculated with an aliquot of washed cells; u, inoculated with an aliquot of non-washed cells), yeast extract concentration (**c**,**d**) (1 g L^−1^; ce, 6 g L^−1^), and extra vitamins and trace minerals (**e**,**f**) (v, supplemented; ce not supplemented) on the evolution of total biomass concentration (*xt*), lipid-free biomass concentration (*xlf*), and residual glucose concentration (***g***) (**a**,**c**,**e**) and the contents of CoQ_9_ (Q9) and CoQ_10_ (Q10) in the total biomass (**b**,**d**,**f**) of *Thraustochytrium* sp. RT2316-16 during the incubation (15 °C, 150 rpm).

**Figure 2 microorganisms-12-01428-f002:**
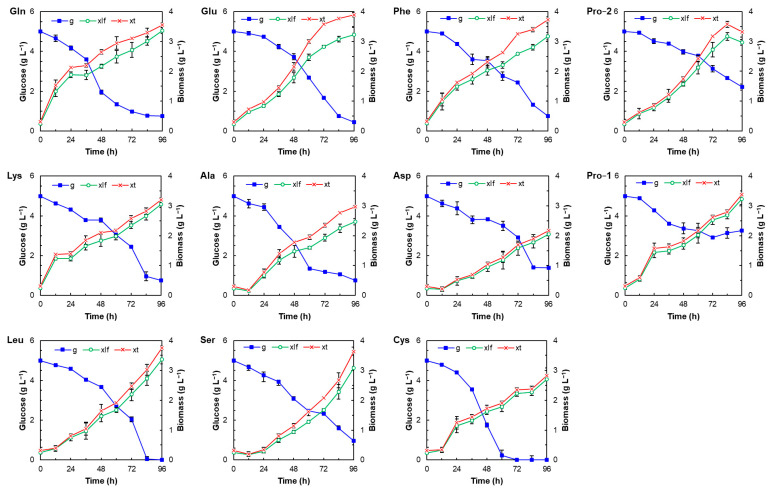
Effect of the amino acid used as the sole nitrogen source for growing *Thraustochytrium* sp. RT2316-16 on the evolution of the total biomass concentration (xt; red line ×), lipid-free biomass concentration (xlf; green line ◦), and residual glucose concentration (g; blue line ▪). Incubation conditions were 15 °C and 150 rpm. The amino acids tested were glutamine, Gln; glutamate, Glu; phenylalanine, Phe; (d) proline, Pro−2 (2 g L^−1^); lysine, Lys; alanine, Ala; aspartate, Asp; proline, Pro−1 (1 g L^−1^); leucine, Leu; serine, Ser; and cysteine, Cys.

**Figure 3 microorganisms-12-01428-f003:**
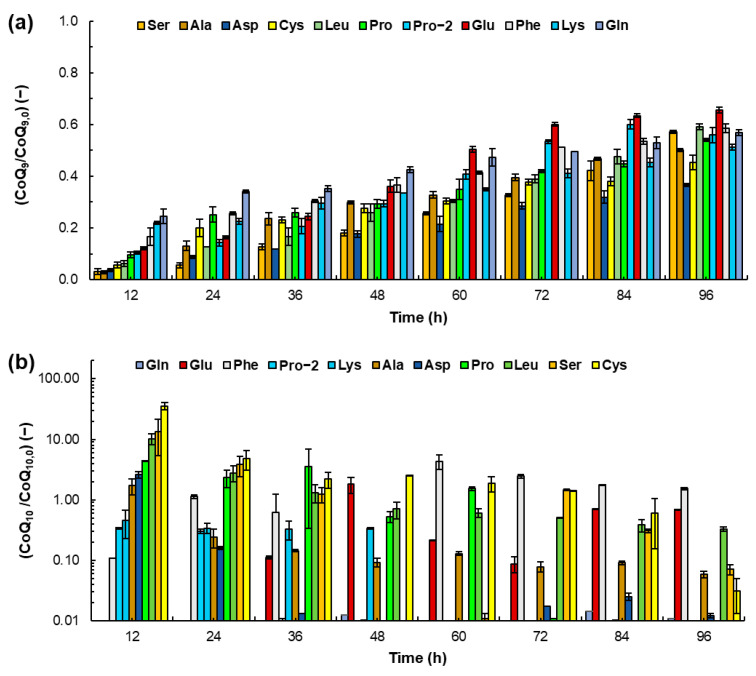
Mass ratio of the content of CoQ_9_ in the total biomass of *Thraustochytrium* sp. RT2316-16 and the content in the initial biomass (CoQ_9,0_) (**a**), and the mass ratio of the content of CoQ_10_ in the total biomass and the content in the initial biomass (CoQ_10,0_) (**b**), at different times. A logarithm vertical axis is used for the CoQ_10_ mass ratio in b for a better visualization. The culture medium contained a single amino acid as the nitrogen source at 1 g L^−1^. Incubation conditions were 15 °C and 150 rpm. The amino acids were: aspartate, Asp; alanine, Ala; serine, Ser; cysteine, Cys; leucine, Leu; proline at 1 g L^−1^, Pro; proline at 2 g L^−1^, Pro−2; glutamate, Glu; phenylalanine, Phe; lysine, Lys; and glutamine, Gln.

**Figure 4 microorganisms-12-01428-f004:**
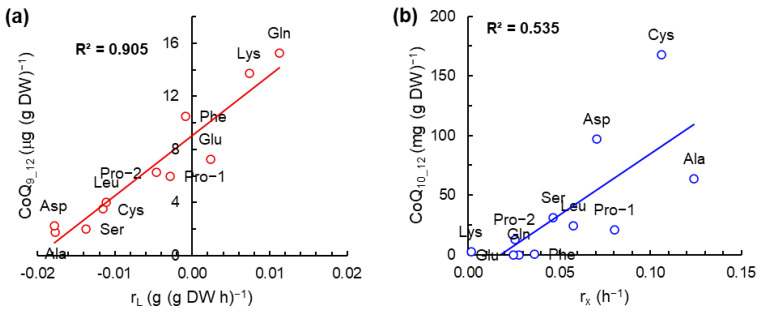
Linear relationship between the CoQ_9_ content in the 12 h total biomass and the specific rate of lipid consumption/synthesis (r_L_) (**a**) and the CoQ_10_ content in the 12 h total biomass and the specific growth rate of total biomass (r_X_) (**b**). The contents of CoQ_9_ and CoQ_10_ were determined in the biomass of *Thraustochytrium* sp. RT2316-16 after 12 h incubation (15 °C, 150 rpm) in media containing a single amino acid as nitrogen source.

**Figure 5 microorganisms-12-01428-f005:**
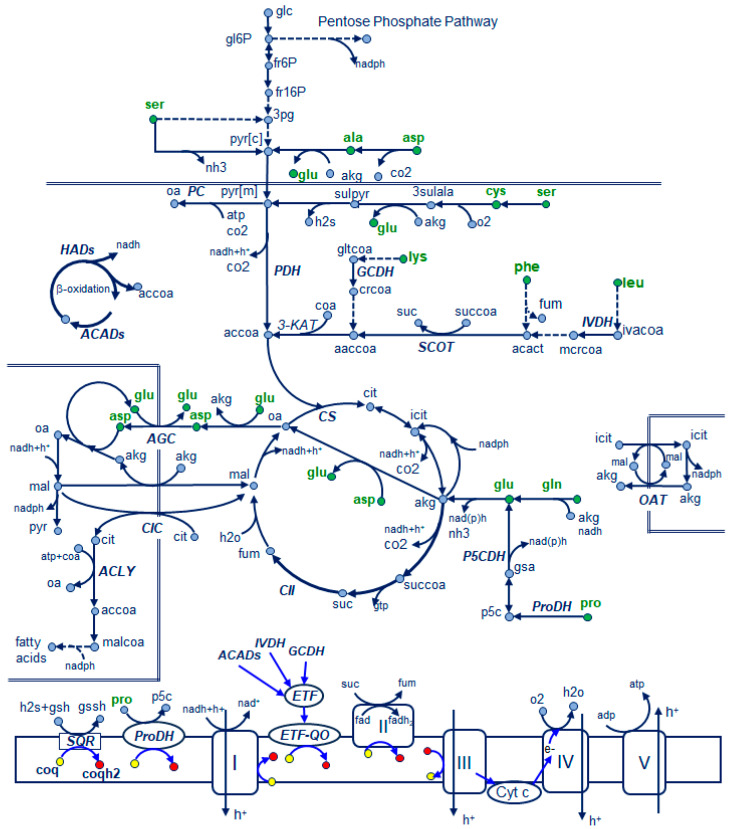
Main reactions in the metabolism of glucose and the amino acids used as the sole nitrogen source for the growth of *Thraustochytrium* sp. RT2316-16 and their involvement in the electron transport chain. Alanine, Ala; aspartate, Asp; serine, Ser; cysteine, Cys; lysine, Lys; phenylalanine, Phe; leucine, Leu; glutamate, Glu; glutamine, Gln; electron transfer flavoprotein, ETF; succinyl-CoA: 3-ketoacid-CoA transferase, SCOT; pyruvate dehydrogenase (DH), PDH; citrate synthase, CS; ProDH proline DH, PCDH; ATP citrate lyase, ACLY; sulfide:quinone reductase, SQR; pyrroline-5-carboxylate (P5C) DH, P5CDH; 3-ketoacyl-CoA thiolase, 3-KAT; aspartate–glutamate carrier, AGC; citrate–malate carrier, CIC; oxaloacetate–malate transporter, OAT; isovaleryl-CoA DH, IVDH; glutaryl-CoA DH, GCDH; acyl-CoA DHs, ACADs; hydroxyacyl-CoA DHs, HADs.

## Data Availability

The raw data supporting the conclusions of this article will be made available by the authors on request.

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
