# Peer review of "Single Amino Acids as Sole Nitrogen Source for the Production of Lipids and Coenzyme Q by Thraustochytrium sp. RT2316-16"

_microorganisms, 2024, doi:10.3390/microorganisms12071428_

Round 1

Reviewer 1 Report

Comments and Suggestions for Authors

Line 53: Include a brief introduction on Thraustochytrids. 

Line 94: include the OD value of the inoculum used for this study.

Line 120: What is the rationale behind testing only proline at 2 g L-1 and and maintaining the pH of the medium containing aspartic at 7?.

2.3.3. Extraction and Quantification of CoQ  - include the reference for this method.

The method explains the lipid extraction. How about CoQ? Is it associated with the intracellular lipids? How about the mitochondrial fraction?

Fig 1. Using c for control and also the figure legend is confusing. 

Lipid-free biomass concentration - Please explain how you calculated this?

In the discussion, the statements in the amino acid studies (lines 294-380) should be supported with relevant references.

What is the purpose of testing individual amino acids for CoQ production? How about the production cost?

Author Response

Responses to reviewers’ comments

 Reviewer 1.

Comment 1. Line 53: Include a brief introduction on Thraustochytrids.

Response: As suggested, in the revised manuscript a brief introduction on thraustochytrids was included. Please see the text in lines 55-60.

Comment 2. Line 94: include the OD value of the inoculum used for this study.

Response: Cell size of thraustochytrid does not allow to use optical density (OD) to estimate biomass concentration because cells rapidly precipitate when the culture is not agitated.

Comment 3. Line 120: What is the rationale behind testing only proline at 2 g L-1 and maintaining the pH of the medium containing aspartic at 7?.

Response: Metabolism of proline produces glutamate and thus similar effects on the biomass production and lipid synthesis were initially expected. Because this was not the case, proline was tested at a higher concentration (2 g L-1) to discard an effect of the concentration. This is explained in lines 378-382.

In the experiment carried out with aspartic acid the initial pH of this culture was 5.3 that did not allow the growth of RT2316-16; this strain can grow in media having pH in the range 6 to 8 (not shown). This explanation can be found in lines 134-135.

Comment 4. 2.3.3. Extraction and Quantification of CoQ  - include the reference for this method. The method explains the lipid extraction. How about CoQ? Is it associated with the intracellular lipids? How about the mitochondrial fraction?

Response: This was our mistake. A reference for the extraction method was included in the revised manuscript. Please see the text in lines 155-156 and the reference in lines 601-603.

The observation of the reviewer is correct, coenzyme-Q is a terpenoid and thus it is found in the hydrophobic portion of membranes. The extraction method used in this work consisted in the sonication of the biomass suspended in methanol for cell disruption (microscopy observation). The lipids released from the biomass solubilize in the methanol and were subsequently recovered with petroleum ether.

In the original manuscript the reference for the quantification method was given. In the revised manuscript this can be found in line 168 and 604-605.

Comment 5. Fig 1. Using c for control and also the figure legend is confusing.

Response: This was our mistake. The letter for identification of the control in Figure 1 was changed in the revised manuscript. Please see the text in lines 195-197.

Comment 6. Lipid-free biomass concentration - Please explain how you calculated this?

Response: In the revised manuscript the relationship used for computing the lipid-free biomass concentration was included. Please see the text in lines 143-146.

Comment 7. In the discussion, the statements in the amino acid studies (lines 294-380) should be supported with relevant references.

Response: References were added in the discussion of the results. In lines 406; 423; 429-431; 443, 444; and 472. The new references can be found in lines 610-611; 621-625; 628-629; and 635-636.

Comment 8. What is the purpose of testing individual amino acids for CoQ production? How about the production cost?

Response: In the revised version a brief introduction was made to give a context to the objectives of the study. In this introduction previous results and the potential use of the results in this study are described. Also, the redaction of the aims of the study were improved. Please see the text in lines 66-80.

Reviewer 2 Report

Comments and Suggestions for Authors

This study focused on evaluating single amino acids as sole nitrogen source to produce coenzyme Q in Thraustochytrium sp. RT2316-16. Overall, this manuscript doesn't seem to have a conclusion, for example the mechanism or the application. And there are sever questions that might be the reviewer’ attention.

1. The tittle was imprecise, since many contents was about the total lipids.

2. Why single amino acid was chosen and why ten amino acids were chosen are not clear.

3. The main purpose of this manuscript was to produce coenzyme Q, why rich medium was not chosen?

4. It's difficult to read the figures 1 and 2, the figure legends were ambiguity.

5. This manuscript is an article about simply descriptive results, no mechanism or a little progress were presented.

6. The conclusion section was too long, and little important information can be understood by readers.

Comments on the Quality of English Language

no

Author Response

Responses to reviewers’ comments

Reviewer 2

Comment 1. This study focused on evaluating single amino acids as sole nitrogen source to produce coenzyme Q in Thraustochytrium sp. RT2316-16. Overall, this manuscript doesn't seem to have a conclusion, for example the mechanism or the application. And there are sever questions that might be the reviewer’ attention.

Response: In the revised version a brief introduction was made to give a context to the objectives of the study. In this introduction previous results and the potential use of the results in this study are described. Also, the redaction of the aims of the study were improved. Please see the text in lines 66-80.

The conclusion section was also revised. Please see the text in lines 480-488.

Comment 2. 1. The tittle was imprecise, since many contents was about the total lipids.

Response: As suggested the title was changed to include the lipids.

Comment 3. 2. Why single amino acid was chosen and why ten amino acids were chosen are not clear.

Response: In the revised version a brief introduction was made to give a context to the objectives of the study. In this introduction previous results and the potential use of the results in this study are described. Please see the text in lines 66-80.

Comment 4. 3. The main purpose of this manuscript was to produce coenzyme Q, why rich medium was not chosen?

Response: We apologize for the poor description of the objectives of our study. The objective of this work was not to produce coenzyme Q. The objective was to study how the thraustochytrid responds to different amino acids in terms of biomass growth, and the content of lipids and coenzyme Q. These results can be used in the design of a chemically defined medium for the production of lipids and/or coenzyme Q. This study should consider the effect of dissolved oxygen.

Comment 5. 4. It's difficult to read the figures 1 and 2, the figure legends were ambiguity.

Response: In the revised manuscript legend of Figure 1 was changed to avoid ambiguity (lines 195-197). Figure 2 and its legend were also modified (lines 279-281).

Comment 6. 5. This manuscript is an article about simply descriptive results, no mechanism or a little progress were presented.

Response: Generally complex nitrogen sources are used to grow these microorganisms. The specific requirements of amino acids by thraustochytrids are unknown. The results in our study can be used in defining a medium for the production of lipids and CoQ10 by related strains.

Comment 7. 6. The conclusion section was too long, and little important information can be understood by readers.

Response: The conclusion section was modified and the main information derived from the experimental study was included. Please see the text in lines 480-488.

Reviewer 3 Report

Comments and Suggestions for Authors

Authors analysed the biosynthesis of total lipids and the content of CoQ9 and CoQ10 in the biomass of Thraustochytrium sp. RT2316-16. As the nitrogen source was used  a single amino acid (ten different aa) at a concentration of 1 g L-1.  The glucose (5 g L-1) was used as the carbon source. Authors determined the biomass concentration,  the content of total lipids and CoQ as a function of the incubation time. Moreover, Authors found that for the aspartate and gluta mate the total biomass was 2.2 ± 0.1 and 3.9 ± 0.1 g L-1, respectively. The content of total lipids in the biomass was highest for media containing glutamate, serine or phenylalanine. The highest content of CoQ9 (39.0 ± 0.7 μg g-1 DW) and CoQ10 (167.4 ± 3.4 mg g-1 DW) was noted  in the biomass when glutamate and cysteine were used as nitrogen source.

Results could be interesting to researchers in the field. Results support conclusions. Methods are correctly described.

The weaker part is the lack of statistical analysis, to show the statistical significacance of results. Following comments should be addressed before the publication:

Line 14-15; Increased as compared to what? Provide information.

Line 53 Define clearly what are thraustochytrids in one-two sentences.

Line 91-94 For what different purposes where used two inoculi? Clearly describe it in material and methods and also results- where was used each inoculum.

Line 117-120 and Fig 5; Correct names of amino acids. Should be as follows: Cys, Lus, Gln, Ala, Glu, Asp, Ser, Leu, Phe, Pro. Correct in the same way in the entire manuscript and Figures.   

Figure 3. On the axis 0Y is CoQ9/CoQ9 and CoQ10/CoQ10. It should be probably CoQ9/ total biomass and CoQ10/ total biomass.

Figure 4. On the axis 0Y is CoQ9_12 and CoQ10-12. In the Figure description Authors write of CoQ9  and CoQ10 t it. Correct it.

General comment: there is no statistical analysis, it is not clear if observed differences are statistically significant. Some statistics should be done, at least in relation to main results presentded in Abstract. Statistical methods should be described in materials and method section.

Comments on the Quality of English Language

Minor editing of English language required.

Author Response

Responses to reviewers’ comments

Reviewer 3

Comment 1. Results could be interesting to researchers in the field. Results support conclusions. Methods are correctly described. The weaker part is the lack of statistical analysis, to show the statistical significance of results.

Response: Thank you very much for this comment. In the revised manuscript the experimental data (concentrations of total biomass, lipid-free biomass, and glucose, and the total lipid content of the biomass) obtained with the different amino acids was compared with data of the control experiment in which yeast extract was used. For this purpose, the method proposed by Hristova and WimleyI (2023) was used. This method is based on the Chi-squared test for goodness of fit with an essential correction for the deviation from normality that arises with small to moderate sample sizes. Please see the text in lines 175-180.

The results of the statistical analysis are presented in the result section in lines: 202-203; 208-209; 218-220; 224-226; 241-242; 269-275; 286-288; 292-294.

Comment 2. Line 14-15; Increased as compared to what? Provide information.

Response: The text in these lines was modified as follow: The biomass grew in media containing glutamate, serine or phenylalanine reached a content of total lipids higher than 20% of the cell dry weight (DW) after 72, 60 and 72 h of incubation. Please see the text in lines 15-17.

Comment 3. Line 53 Define clearly what are thraustochytrids in one-two sentences.

Response: As suggested, in the revised manuscript a brief introduction on thraustochytrids was included. Please see the text in lines 55-60.

Comment 4. Line 91-94 For what different purposes where used two inoculi? Clearly describe it in material and methods and also results- where was used each inoculum.

Response: The strain is kept at -18°C and we need it to be active for rapid growth after transferring to a new medium. This was explained in lines 104-106.

Comment 5. Line 117-120 and Fig 5; Correct names of amino acids. Should be as follows: Cys, Lus, Gln, Ala, Glu, Asp, Ser, Leu, Phe, Pro. Correct in the same way in the entire manuscript and Figures.  

Response: The text was changed in lines 131-133; 395-396.

Comment 6. Figure 3. On the axis 0Y is CoQ9/CoQ9 and CoQ10/CoQ10. It should be probably CoQ9/ total biomass and CoQ10/ total biomass.

Response: This was our mistake. In the original manuscript we did not define what was denominator (CoQ9,0) in the ratio shown in Figure 3a. The content of coenzyme Q9 in the biomass in the inoculum (time 0 h) (CoQ9,0) was used to normalize data (lines 303-304).

Comment 7. Figure 4. On the axis 0Y is CoQ9_12 and CoQ10-12. In the Figure description Authors write of CoQ9  and CoQ10 t it. Correct it.

Response: This was our mistake. In the original manuscript we did not define what was denominator (CoQ10,0) in the ratio shown in Figure 3a. The content of coenzyme Q10 in the biomass in the inoculum (time 0 h) (CoQ10,0) was used to normalize data (lines 303-304).

Comment 8. General comment: there is no statistical analysis, it is not clear if observed differences are statistically significant. Some statistics should be done, at least in relation to main results presentded in Abstract. Statistical methods should be described in materials and method section.

Response: Please see our response to Comment 1.

Round 2

Reviewer 2 Report

Comments and Suggestions for Authors

All of my issues has been addressed.

Comments on the Quality of English Language

no